# Insights about Screen-Use Conflict from Discussions between Mothers and Pre-Adolescents: A Thematic Analysis

**DOI:** 10.3390/ijerph18094686

**Published:** 2021-04-28

**Authors:** Kathleen Francis, Hanneke Scholten, Isabela Granic, Jessica Lougheed, Tom Hollenstein

**Affiliations:** 1Behavioral Science Institute, Radboud University, 6525 XZ Nijmegen, The Netherlands; h.scholten@pwo.ru.nl (H.S.); i.granic@pwo.ru.nl (I.G.); 2Technology, Human and Institutional Behavior Group, Department of Communication Science, University of Twente, 7522 NB Enschede, The Netherlands; 3Department of Psychology, University of British Columbia, Kelowna, BC V6T 1Z4, Canada; jessica.lougheed@ubc.ca; 4Department of Psychology, Queens University, Kingston, ON K7L 3N6, Canada; tom.hollenstein@queensu.ca

**Keywords:** digital media, screen use, pre-adolescence, parent–child conflict, thematic analysis

## Abstract

Digital screens have become an integral part of everyday life. In the wake of the digital swell, pre-adolescents and their parents are learning to navigate seemingly new terrain regarding digital media use. The present study aimed to investigate parent and pre-adolescent perceptions of screen use and the source of conflict surrounding digital media. We employed a qualitative thematic analysis of 200 parent and pre-adolescent dyads discussing screen use. Our analysis showed five overarching themes for screen use perceptions and conflict: screen time, effects of screen use, balance, rules, and reasons for screen use. In contrast to previous studies that mainly focused on parental perceptions, we were also able to shed light on pre-adolescent perceptions of screen use and the difference in opinions with their parents. Furthermore, we found that patterns of the source of screen use conflict were oftentimes rooted in the age-old developmental tug of war between autonomy-seeking pre-adolescents and authority-seeking parents. Though navigating autonomy-granting and seeking behavior is familiar to developmental scientists, negotiating these challenges in a new digital world is unfamiliar. Autonomy support, open dialogue, and playful interaction between parents and children are needed to understand and resolve conflict of digital media use in family contexts.

## 1. Introduction

Conflicts between parents and children tend to increase in frequency at the transition to adolescence. Quarrels and bickering over everyday issues are commonplace in parent–adolescent relationships [1]. Pre-adolescents begin to challenge the rules and norms put in place by parents, and they seek new modes of independence [2,3]. Young people are becoming independent and engaging in the social world—they are still dependent on caregivers in some key domains while simultaneously building their own voices and opinions. Early and pre-adolescence are marked by an increase in negative emotional states as children transition and move to the next developmental phase of life [4]. In turn, parents perceive adolescence as the most difficult time of childrearing [5]. However, researchers agree that conflict in this transitional period is normative and functional in transforming family relationships [2]. For parents and pre-adolescent children (11–12 years old) in today’s digital world, conflicts about technology and screen use pose a new and escalating challenge for families [6]. Screen-use conflicts take many forms including the appropriate age to get a smartphone; signing up for social media accounts such as Facebook, Instagram, and Snapchat; and the appropriate use of screen devices [7].

Screen-use conflicts are often driven by caregiver’s fear of the potential negative effects of screen use. Such anxiety is amplified by media coverage and popular science publications, which portray digital devices as harmful to healthy development and growth [8,9,10] despite little to no scientific evidence to support these fears [11]. Though hundreds of studies have been conducted on screen use and screen time [12], the literature remains inconclusive. However, headlines in the media regularly suggest that screen time causes a host of negative mental and physical health outcomes [13]. Consequently, parents fear that screen use may degrade mental well-being [14], increase aggression [14], and lead their children to become addicted to screen time [14]. Therefore, studies suggest that many parents view screen time as largely negative.

In contrast to parents’ opinions about screen time, we know relatively little about pre-adolescents’ perceptions of screen use. The scarce literature that is available suggests that young people view screen use as an opportunity to connect with their peers, assert their autonomy, and learn about the world [15]. Despite the insights from studies of parents’ concerns and the scarce literature of youth’s perceptions of screen use, no one to our knowledge has systematically studied screen-use conflict in naturalistic conversations. Current attempts to understand both parent and pre-adolescent perspectives and screen use have been shallow, focusing on mere time metrics (e.g., how many hours spent on screens a day) and reliant on self-report measures that have been shown to either be poorly related or completely unrelated to screen use [16,17]. Inconsistent conclusions and poor statistical methods render a call to action for a different methodological approach to understanding screen-use phenomena [13].

In particular, qualitative analyses of naturalistic conversation can offer tremendous insight into pre-adolescent perspectives and parent–child screen-use conflicts for two reasons. First, it is important to know what youth actually think about screen use. Doing so will help us understand pre-adolescents’ perspectives, feelings, and motivations. Second, by better understanding young people’s views on screen use, we are better able to investigate parent–child conflicts about screen use. Currently, pre-adolescent views have been absent from screen-use conflict discussions. However, one possibility is that conflicts arise because youth think about and use screens differently than parents. Learning about youth perspectives also helps to inform our understanding of parent and pre-adolescent screen-use conflicts. Thus, the following questions remain: how do conversations about screen use unfold between parents and children, and what are the core themes of screen-use conflict? In the current study, we examined screen use conflicts between pre-adolescent (11–12 years old) children and their mothers using qualitative thematic analysis. We analyzed both parent and child perceptions to gain insight into their views of screen use. This allowed us to examine screen-use conflict within real-world parent–child interactions, thereby enabling us to identify both unique and familiar aspects of parent–child conflicts in the context of screen time use.

## 2. Materials and Methods

### 2.1. Participants

The present study used data from a longitudinal study conducted in a small city in Canada. Mother–child dyads (N = 200) were recruited from a database of families interested in participating in research. The mean age of the children was 11.49 (SD = 0.55; range = 10–13). Of the 200 dyads that participated in the primary study, 113 dyads discussed screen time as one of their primary sources of conflict. Of these 113 dyads, 22 dyads were excluded due to missing data (*n* = 2), incoherent voices (too low; *n* = 2), or because the data did not provide relevant information for our research questions (i.e., screen use was not discussed) (*n* = 17). This resulted in a final sample of 91 mother–child dyads (*n*_boy_ = 50; *n*_girl_ = 41).

### 2.2. Procedure

Participants completed questionnaires on a computer in the lab prior to engaging in a 4-min videotaped conflict discussion. The conflict topic was selected from one of the questionnaires for which mothers and children identified the top 3 conflict topics they argue about. The researchers chose the topic rated as the highest intensity from those two lists for the dyad to discuss. In the present study, we only analyzed transcribed dyadic data in which screen use was discussed.

### 2.3. Data Analysis Strategy

To analyze the screen-time conflicts between parents and pre-adolescents, we used qualitative thematic analysis [18]. Specifically, the first and second authors separately analyzed the data following Braun and Clarke’s (2006) recommendations. We proceeded with five steps. First, we read and re-read all transcripts to familiarize ourselves with the data and briefly discussed our initial interpretations. Second, we generated codes by randomly selecting 20 transcripts (see Table 1 for an example). Next, we conferred on initial codes and then selected and coded 20 new random transcripts using the codes from the first set. This procedure was repeated for the third and fourth sets of 20 random transcripts, applying all previously generated codes to a subsequent set. After going through this procedure with a total of 80 transcripts, no new codes emerged and we therefore stopped coding, following standard procedure [18]. Third, we evaluated generated codes, and we named and defined overarching themes. Fourth, the authors met for an analysis session, which involved discussing, comparing, and verifying the themes generated by each author. Specifically, themes were verified by cross-referencing the thematic scheme with the transcripts to validate the themes. As such, we re-read the transcripts and referenced the thematic scheme while reading. Finally, the themes were reviewed by two external readers to validate the credibility of the proposed themes.

## 3. Results

### 3.1. Perceptions of Screen Use

In the early stage of the analysis, we found 58 unique codes—29 for parents and 29 for children. Following our data analytic strategy, we first organized these codes into themes. We identified a total of five themes and various sub-themes. These themes describe the full spectrum of the perceptions of the parent and pre-adolescent screen-use conflict. Themes were (1) screen time (e.g., number of hours and times of use per day), (2) the effects of screen use with the sub-themes of (a) positive effects and (b) negative effects, (3) balance with other tasks including sub-themes of (a) good balance and (b) screen time instead of other tasks, (4) rules, and (5) reasons to use screens. All five themes were the same for both parents and children, but the sub-themes differed between parent and pre-adolescents. A comprehensive list of all themes and subthemes differentiated by parents and pre-adolescents is shown in Table 2. The first column contains parent themes. The second column contains pre-adolescent themes. Headers with bolded text represent themes. Underneath each theme is a bulleted list of subthemes. Below, we separately discuss all themes and subthemes for parents and children.

### 3.2. Parental Perceptions of Screen Use

Screen Time: One central theme of screen-use discussion is the amount of time spent on devices. In the sub-theme of too much time per child, parents were generally worried about the amount of time their children spent on screens both for school and recreational use. Parents largely commented that their children spend too much time in front of screens. In addition, parents noted that they wanted their children to initiate a decrease in the amount of time they spend using screens, saying things such as “I would like you [pre-adolescent] to stop watching your screen yourself instead of [me] asking you to”. Moreover, parents often referenced that their children could not accurately track the time they spent using screens (e.g., pre-adolescents believe they had been on the iPad for ten minutes while it was actually 30 min). Parents also referenced their inability to teach their child(ren) how to better spend their time and referred to children who use screens as lazy. The parental language of how to better spend time indicated that screen use is, in their opinion, not a valid or productive way for a child to spend time. Generally, parents perceived screen use as inappropriate use of time and argued that their children should be doing something more productive, saying “You have other things you could do but you waste your time on the iPad”. 

Effects of Screen Use: The effects of screen use were a main theme, with both positive and negative effects being reported. In terms of positive effects, many parents acknowledged the positive effects screen use can have on their children. Parents identified three main areas influenced by screen use—socio-emotional development, family dynamics, and educational/academic. For socio-emotional development, many parents recognized the importance of screen use as a way to communicate with friends. Though they often referred to screen use as inferior to offline communication, they admitted that it was the way youth interact and thus may have some benefits to social development. Moreover, screen use was referenced as a family activity, as parents perceived screens as an opportunity to play games or watch digital content as a family, thereby improving family dynamics. Finally, parents recognized the positive educational effects of screen use. Parents generally praised screen use when referring to academic outcomes such as learning in the school setting.

In the negative effects of the screen use sub-theme, interference with healthy development was a common concern. Parents construed screen use in all forms (e.g., mobile phones, tablets, and video games) as a source of disruption in the healthy development of their child(ren). Parents identified multiple areas that are influenced by screen use including social/emotional development, physical development, cognitive development, and moral development. First, effects on appropriate social/emotional development, which refers to the frequent reference of the parents’ fear that screen use may cause cyberbullying, social isolation, and less face-to-face socialization. For example, parents referred to screen use as causing less motivation to leave the house. Second, effects on physical development included concerns from parents about how screen use interfered with time spent outside, that it led to an absence of interest in physical activity, and that it led to a disruption of sleep. For example, one parent brought up concerns of blue light causing headaches, saying “Do you think it’s causing you headaches? Looking at a screen at night?” The third was the effect of screens on cognitive development, which refers to the regular references of parents’ that the use of screens is cognitively taxing, leaving no time to truly relax. In addition, there was a pattern of parents noting that screen use stunts creativity and reduces imaginative thinking. For example, one parent said “Video games take away your imagination, you don’t need to think, they are rotting your brain”. The final area within the negative effects of screen use was moral development, which refers to the frequent reference of the parents’ association of screen use as the source of lying. Parents brought up accounts of their child(ren) lying about their screen use, the content of their screen use, and sneaking devices. For example, parents said things such as “Yeah, I think you sneak it a lot too. I think you tell me you’re going to bed and then you listen for me to come upstairs”. In addition, there was a pattern of concern regarding the violent and potentially inappropriate content of video games, YouTube videos, and other online spaces. Parents made numerous references to the predatory nature of video games and screen applications, including their concern that they are designed to be addictive and destructive to children’s health.

Balance: An overarching theme in the parental perceptions of screen use was the balance between screen use and other tasks. The sub-theme of spending time on screens instead of something else refers to the common complaint made by parents that screen use prevents their child from completing necessary tasks such as schoolwork and chores. In addition, they noted that screen use also took precedent over all other tasks such as reading, playing an instrument, or learning a new skill. Parents often commented that “So you should be outside doing things or reading a book, things that expand your mind like playing the piano” or “You used to read all the time before you had a phone” On the other hand, the sub-theme of good balance refers to a pattern of parents noting there was a balance between screen time and tasks such as homework completion, saying things such as “I don’t mind when you play on the computer when you get your other things done, which you usually do”. Though this was not the dominant perspective, many parents still noted that balance was present due to using screen time as a negotiating token.

Rules: Rules surrounding screen use were also a common theme including sub-themes of frustration (regarding not following rules) and the necessity of rules. Parents were often frustrated and concerned with how frequently their children disregarded the set rules regarding screen use. In most discussions, parents mentioned things such as “So, yeah, that is definitely something that annoys me to no end that you can’t turn off immediately when I say hey, turn it off”. Though there was often a mutual (parent and child) understanding of the necessity of rules surrounding screen use, with pre-adolescents noting things such as “Yeah, I think it is okay. Limit is okay”. Parental frustration was more common.

Reasons for Screen Use: An important theme that arose was the reason for screen use. In the sub-theme of acceptance, parents’ general disposition was that they did not understand why their pre-adolescent is so keen to play and interact with their screen. They often referenced a generational gap as the reasoning for their lack of understanding, noting “It’s hard because it was never a part of my life really growing up. So I don’t really understand the like, I don’t… I can’t empathize with you, the whole, you know, wanting to do it so bad” or “We had TV but that’s it”. Though some parents used their lack of understanding as a justification for less screen use for their young adolescent, others chose to accept that they will not understand and instead tried to embrace the new digital landscape, saying things such as “I think I’ve come to a place of accepting…that this is just a part of the world that you guys live in”. In the social sub-theme, there was an overall parental consideration that screens facilitate social communication and connection, and that these are some of their primary uses.

### 3.3. Pre-Adolescent Perceptions of Screen Use

Screen Time: In contrast to parents, in the sub-theme of “screen time is fine”, child pre-adolescent participants tended to not perceive the amount of time they spent online and using screens as problematic. In other words, although they acknowledged the large amount of time using screens, they did not think it was too much. Interestingly, in the sub-theme of parent screen time, pre-adolescent participants referenced their parents’ hypocrisy in screen time and screen use. For example, they noted that their parents used screens as or more often as they did, and their parents would not engage with them if they were using digital devices. In one discussion, a pre-adolescent said “You still don’t love me and you don’t give me enough attention. You spend more time on the laptop than with me. And it’s not just, and it’s not just now. It’s before, you use your laptop and phone”.

Effects of Screen Use: The effects of screen use were a main theme, with both positive and negative effects as sub-themes. In the positive effects of screen-use sub-theme, many pre-adolescents acknowledged the positive effects of screen use. The social and emotional sub-themes focused on how pre-adolescents perceived the use of digital devices and screens as adding value to their lives and facilitating meaningful connections, often referencing the collaborative nature of online games. They noted screens helped them to feel more connected to the world. The sub-theme of fun and engagement focused on the importance, ease, and joy of using digital devices to communicate and socialize with their friends. In fact, pre-adolescent participants suggested they preferred online over in-person interaction because it feels safer and are in more control of “how” and “when” they socialize. Finally, the sense of control/safety sub-theme focused on pre-adolescents’ desire for online over in-person communication. The desire to communicate online instead of offline suggests there is a sense of safety in their belongingness online, saying they were “comfortable in [their] little bubble”.

Pre-adolescents noted few negative aspects of screen time. For the family dynamics sub-theme, pre-adolescents shared their parent’s concern about screen dependency and viewed screen time as a source of interference in familial dynamics. For example, they referenced fighting with siblings over the use of devices, differences in sibling privilege in screen time, and fighting with parents about screen time. In one case, a pre-adolescent participant said “We fight a lot about this [screen time], and I don’t like when that happens”.

Balance: The balance of screen use was a main theme, with good balance as a sub-theme. In the sub-theme of good balance, pre-adolescents often thought they were doing a good job of completing other necessary tasks and “earned” their screen use. In response to comments from parents about doing other activities such as reading or engaging in thought-provoking activities such as learning a new skill, they mentioned how video games or online activities include a lot of reading and problem solving. In addition, pre-adolescents were often open to negotiating screen time for time spent on other things such as reading or chores.

Rules: The balance of screen use was a main theme, with frustration regarding a lack of flexibility in rules, parent not following rules, and the necessity of rules as sub-themes. In the sub-theme of frustration (regarding a lack of flexibility in rules), pre-adolescents often mentioned frustration regarding screen use rules, similar to what parents noted. However, they commented on the lack of flexibility from parents in their screen-time rules and noted they did not feel like they had any say in how the rules were negotiated. Pre-adolescents felt their voice and concerns were not taken into consideration when setting up screen-time boundaries. Furthermore, in the sub-theme of the parent not following rules, pre-adolescents referenced the hypocrisy that their parents did not have to follow the same screen-time rules that were implemented for the child. They noted things such as “Well, you [parent] and dad are on your phone at the table all the time and I can’t be, which, like, isn’t fair”. Interestingly, in the sub-theme of the necessity of rules, pre-adolescent understood and recognized why screen use rules were necessary and were not upset that baseline rules were in place, saying things such as “Turning it off at night is good for me so I don’t get notifications”.

Reasons for Screen Use: Reasons for screen use was a main theme with recreational, normality, social, and educational sub-themes. In the recreational sub-theme, pre-adolescents often referred to screen use as recreational and noted that screen use was something to do when bored or when there is nothing else to do. In the social sub-theme, pre-adolescents referenced screen use as a way to communicate and connect with their friends either within a game setting or just chatting. For example, one pre-adolescent summed it up succinctly, saying “They’re fun. It’s a fun way to connect with your friends. Have fun with them even when you can’t like be with them in person”. In addition, in the educational sub-theme, there was a strong pattern of pre-adolescents referencing the educational uses of screens both inside and outside of school, saying things such as “Uhm, sometimes it’s like educational, like, uhm, game theory that teaches you a lot of things, like, for example, like, it teaches us about math like for like numbers like the death counts in Disney use math. Like, I learned how much land that a lion needs to feed itself”. Finally, in the sub-theme of normality, pre-adolescent participants frequently referenced the normalcy of screens in their lives. They noted that screen use is simply a part of the natural order of the world and they could not imagine a world without screen technology or how they would function in such a world. They often mentioned how commonplace video games, mobile phones, and tablet devices were in their peer groups. Pre-adolescent participants also spoke about the lack of understanding from their parents about why they enjoy using screens. They referenced parents not understanding the depth of their socialization or the joy their online lives bring them. They noted this comes from a generational difference in screen use. Pre-adolescents often reported that their parents did not know how to use devices, nor did they understand video games or the nuances of online communication. They also referred to devices as a source of entertainment and a better alternative to “less fun” activities like reading books. In addition, they referred to their parents as “old fashioned” for their lack of understanding. 

## 4. Discussion

In the current study, we examined screen-use conflicts between parents and pre-adolescents using qualitative thematic analysis to investigate two gaps in our knowledge base regarding screen use. First, we observed real-world parent–child interactions to get a better understanding of the parental, but especially pre-adolescent, perceptions of screen use, since the latter are almost fully missing in screen-use literature. Second, we aimed to better understand the gap between the screen-use perceptions of parents and their children to start disentangling the source of conflict in screen-use discussions. We found that parents and children were divergent in their perceptions of screen use. Though similar themes arose, the pre-adolescents’ values and attitudes toward screen use were oftentimes in opposition with their parents. Parents mostly reported that their children spent too much time on their devices, had problems with following screen-time rules, and were not able to find a balance between screen use and other tasks. Though parents reported both positive and negative consequences of their children’s screen use, their perceptions were mostly focused on the negative. Parents acknowledged that technology could be useful for education and socialization. However, they mostly saw it as a threat to healthy social, cognitive, and moral development, as well as in direct opposition with their child’s ability to be productive. Children generally reported that their amount of screen time was fine for them or could be increased, were frustrated about the inflexibility of their parents regarding screen-time rules, and were mostly able to balance screen time with other activities and tasks. In contrast to their parents, pre-adolescent perspectives did not view screen use as threatening or worrisome. They focused on rebutting their parents’ concerns by emphasizing the delight screen activities brought them socially, creatively, and cognitively.

Our first aim was to bridge the knowledge gap regarding pre-adolescent perceptions of screen use through observing screen-use discussions between parents and their children. We found that pre-adolescents perceived screen use as a joyful and often social activity. For example, they perceived screen use as natural, valuable, recreational, and essential to their ability to socialize and explore their interests. They were not worried about the amount of time spent using screens and instead often expressed wanting to have even more screen-time. Though they recognized and respected the need for boundaries around screen use, they felt their opinions and reasoning for flexibility in screen-use rules were lost in parents’ misunderstanding of the role screens play in their lives. In contrast to their parents, pre-adolescents generally had more positive views of screen time; however, they also acknowledged that screen use is a source of interference in family dynamics, such as fighting over devices with siblings or arguing with parents about screen use.

Though past literature on pre-adolescent perceptions of screen use has been limited, the themes we identified in the current study were consistent with past findings [19]. In general, children seemed to know about the potential negative effects of screen use but experience the wonders of their digital world and have a more nuanced understanding of what screens are used for and how to interact with them. They saw screen use as natural to their development and not disruptive to their productivity. However, it is important to note that capturing the child’s perspective of screen use was difficult compared to evaluating parental perception. Given the unstructured nature of the conversations between parents and children, the parent often dominated the conversation and guided the child’s answers to align with their perceptions, leaving less opportunity for children to express their perceptions without bias. In addition, pre-adolescents were often quiet and simply did not engage in the conversation. Thus, although we got an impression of the pre-adolescent perceptions, our overall observation was that many children withdrew from the discussion, thereby limiting our capacity to capture the child’s perspective.

Our second aim was to bridge the gap between parental and pre-adolescent perceptions of screen use to better understand where screen-use conflict originated from and how these conflicts might be prevented in the future. In addressing this question, we identified two patterns within the analysis of our themes and sub-themes. First, screen-time conflicts could be partly explained from a developmental perspective: much of the conflict discussions were focused on the battle between autonomy and authority between children and parents. On the one hand, parents expressed a strong desire for their children to obey rules put in place for screen time and they became frustrated when children disobeyed. As noted in our results, parental participants frequently referred to their child not ending screen use when asked. On the other hand, children desired to express and direct their own screen time use and felt infringed on when they are not allowed to be autonomous. Specifically, we found that pre-adolescents were aware that having cell phones and screen-time privileges allowed for freedom. In addition, children wanted trust in their knowledge of appropriate digital hygiene and their ability to censor content. Moreover, they often argued that screen time and the use of digital devices did not undermine other obligations such as participating in extracurricular programs, house chores, and homework. The nature of these arguments was indicative of children seeking the autonomy to govern their own time. Though these themes did indeed contain screen-use content, underlying this surface content are age-old themes centered around authority and autonomy. From our view, screen use seems to be a proxy for a typical parent–child relationship dynamic observed within other contexts (e.g., returning home on time and sweets intake) and over many past generations. Such results support prior research that pre-adolescence is characterized by a process of changes toward autonomy and is associated with oppositional interactions between children and parents [20].

In order to deepen our understanding of family dynamics in the digital age, it seems important to remain cognizant of developmental themes such as increases in parent–child conflict that have long typified this developmental transition. It may also be important to integrate theory and new insights about perceptual and interpretation differences around screen time to build guidelines that help parents and children better navigate conflicts in the digital age. For example, self-determination theory (SDT) could be useful for understanding screen-time conflicts. SDT posits that the motivation to self-regulate or change behavior can be sparked either internally or externally (e.g., reward and punishment). Accordingly, when parents foster their children’s internal motivations (through autonomy support), they are bolstering their child’s innate natural process to become intrinsically motivated to be intentional, internalize, and maintain change toward healthier behaviors and development [21]. More specifically, parents can help build behavior change through autonomy support by providing structure in a democratic manner, which respects children’s interests and feelings and builds trust [21,22]. With respect to screen use, SDT suggests parents should start a dialogue with their children that is open and interested in their children’s thoughts and feelings about screen use and time management.

In addition to the insight of a developmental lens, a second pattern that arose in conflict discussion is that it seems to be the case that today’s parents are faced with an experiential gap with screen-based technologies—they have never lived in a world where social connections, entertainment, and school/education exist at the hybrid boundary of in-person interactions and on screens. Though there have always been generational differences in technology and media (e.g., rock-and-roll, radio, and television), screen use appears to be a new challenge. Children perceive screen use as a socialization tool that provides meaningful connections. In addition, they thought of screen use as space to exercise their creativity and problem-solving skills. Finally, they consider screens as an extension of their offline lives, which has been referred to as hybrid reality: the constant interaction between online and offline spaces without clear boundaries [13].

In contrast, most parents noted that video games do not have real-world relevance and that social interaction on screens is less “real” compared to face-to-face interaction. That is, parents feel socialization on screens is inferior to real-world interactions and thus do not recognize the concept of hybrid reality themselves. This notion undermines the reality of the pre-adolescents’ perception that their online socialization is meaningful while simultaneously confirming parents do not perceive screens as a part of “real” life. By missing the constantly interacting offline and online worlds of their children, parents are unable to fully grasp the experience of their child. By investing time to understand the blended online and in-person experiences of young people through open dialogue and playful digital interaction, parents would be better able to grasp the multi-faceted identity of their children’s hybrid reality.

### Limitations and Recommendations

While this study presented unique and novel findings, its limitations must be noted. First, people are unreliable when recalling details about their use of social media or other digital technologies, to which pre-adolescents are especially prone [16,23]. Thus, future research should be designed to capture pre-adolescent perceptions without the burden of parental interruption and provide support for accurate recall. For example, researchers could utilize structured stimulated recall interviews [24] to better capture children’s perceptions and gain more insight into the discrepancies of screen-specific parent–child perceptions. Further, structured interviews with specific questions regarding screen use could provide insightful information. Moreover, our discussions invited only the mother as the representative of the parents to discuss with the children. As some of the children mentioned they enjoy playing video games with the father, the discussion dynamic might have been different if it was a father–child interview or an interview with other primary care-taker figures. For example, fathers often engage in more play behavior than mothers, which could lead to different conversations regarding screen use [25]. In the future, it would be useful to investigate interactions with other caregivers regarding screen-time discussion.

## 5. Conclusions

In general, we found that parents and children have conflicts around screens, and they hold different perspectives and place different values on screen use. Our study confirms many parent perceptions that previous research has found, and we contribute to the pre-adolescent perspective that has been lacking in previous research. Most importantly, young people are more open to how screens can be used for a wide variety of life domains but are also willing to compromise about screen use with their parents. Despite the differences between parents and children, they all recognize the potential costs and benefits of screen use. Importantly, while navigating screen-use conflict, they are simultaneously navigating normative developmental processes. During the digital age, youth have not evolved to be entirely different beings, nor has the parent–child relationship transformed as a result of digital interaction [26]. Children will always seek, and have always sought, love, validation, and safety from their primary caretakers while gaining their autonomy [20], whether the conversation is about the use of the new family radio or more time on the gaming console. Conflict and tension are a natural part of family dynamics, especially during puberty and family transition [27].

However, the current technological landscape—in which offline and online worlds are integrated and hard to disentangle, especially for young people—poses a unique challenge in human history. Thus, there are both familiar and novel challenges that parents and children will need to navigate within our new digital ecosystems. In order to address these challenges, we need to change the social messaging and media discussion around screen use, and we need to normalize the use of screens. Parents themselves seem to experience challenges in their relationships to screens and may need to consider the impact of their screen-use messaging to their children. Specifically, autonomy support by democratically providing structure, open dialogue, and playful interaction between parents and children are necessary to help families improve relationships and resolve conflict as digital devices continue to shape our social world.

## Figures and Tables

**Table 1 ijerph-18-04686-t001:** (**a**) Example of coded transcript for parental perceptions. (**b**) Example of coded transcript for pre-adolescent perceptions. (**c**) Example of coded transcript for source of conflict.

(**a**)	
Transcript	Codes
Parent: It’s really addicting. It’s hard… it’s hard to understand the fine line between having a little bit of fun, but then knowing when to stop or… How do you feel about that? Do you think you use it the right amount or do you think it’s ok?	Screen time is addictive Difficult to have fun but not turning into an addiction
Child: I think TV is in right amount. But maybe screen time, a little bit more, like YouTube.	
Parent: …It’s damaging your eyes and makes it harder for you to concentrate accurate… We’ve talked about that before. Right.	Damage to eyes Negatively affect concentration
Child: Yeah.	
Parent: …Even if you don’t have homework, you shouldn’t spend the whole time in it.	Should not spend too much time on screen
(**b**)	
**Transcript**	**Codes**
Parent: So we have to talk about playing video games. Child: video games are fun you can do lots of family things with them but they’re kind of over addictive sometimes and can kind of spoil kids.	Video games are fun Video games sometimes can be addictive Video games can spoil kids
…	
Child: They’re fun. It’s a fun way to connect with your friends. Have fun.	Screen use is a way to connect with friends
(**c**)	
**Transcript**	**Codes**
Parent: I don’t mind you playing video games. I just don’t want them to take up so much time that… You won’t do anything else but play video games or you can really forget about homework or things that you should be doing because you need to play video games.	Child forgot about homework because of video games
… Child: Sometimes there go with the arguments. Parent: Why? Child: Because I don’t want to leave screen time here, but you want to make me stop.	Child did not obey the parent to stop using screen time

**Table 2 ijerph-18-04686-t002:** Parental and pre-adolescent perceptions of screen-use.

Themes (Bold) and Subthemes (Italics)	
PARENT	PRE-ADOLESCENTS
**Screen Time**	**Screen Time**
Too much screen time	Screen time is fine Parent screen time
**Effects of Screen Use**	**Effects of Screen Use**
Negative: - socio-emotional - physical - cognitive - moral Positive: - socio-emotional - family dynamics - educational	Negative: - moral - family dynamics Positive: - socio-emotional - educational - fun and engagement - sense of control and safety
**Reasons for Screen Use**	**Reasons for Screen Use**
Acceptance of screen use Social	Recreational Normality Social Educational
**Rules**	**Rules**
Frustration: not following rules	Frustration: parents not following rules Frustration: not enough flexibility
**Balance**	**Balance**
Spending time on screen instead of something else Good balance	Good Balance

## Data Availability

The data presented in this study are available on request from the corresponding author. The data are not publicly available because this is an ongoing five-year longitudinal study.

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
