# Peer review of "Insights about Screen-Use Conflict from Discussions between Mothers and Pre-Adolescents: A Thematic Analysis"

_ijerph, 2021, doi:10.3390/ijerph18094686_

Round 1

Reviewer 1 Report

Authors address an important topic: mothers’ and pre-adolescents’ mutual perceptions of their communicative and potentially conflictual interactional exchanges about screen overusing.

Strengths are two: a) employing a qualitative thematic analysis that brought out the “five overarching themes”; b) filling a gap, that is, bringing out pre-adolescents’ point of views (i.e., meanings and motivations that they attribute to screen using and overusing).

However, before article is published two minor revisions are necessary:

First, in Results 3.1. section the description of themes and sub-themes is not clear and difficult to understand. A direct reference to Table 2 and how to read it might perhaps be more useful?

Second, why is no quantitative data reported, such as the percentages of themes and sub-themes referred to mothers and children? I would at least mention theme and sub-theme prevalences.

Reviewer 2 Report

The authors took up the interesting topic of conflicts between parents and children, which conflicts have been limited to the use of technology and equipment (with "digital screen") quite often used today in some part of the world. In fact, it is still a sphere that is relatively little understood. The findings of the researchers confirmed the presence of "eternal" conflicts between parents and children in modern families, whose "driving force" is the desire of parents to dominate and the need for adolescent autonomy - however, the subject of the conflict is relatively new ("screen - use").

The researchers limited themselves to qualitative analysis, although the collected research material allows for a quantitative analysis (even in accordance with the narrative approach).
The text is written in a way that is accessible to most readers.

Researchers point to the limitations of research related to, inter alia, only the interaction of mother and child. If the material collected allows it, it would make sense to check whether the child whose statements were transcribed was the eldest child in the family, because the eldest child generally "blazes the path" for younger siblings.
I have a question for the authors: Were the results of the conflict research similar in the case when the child is the only one in the family and the second child, etc., regardless of the child's birth order? There is also a question about the family's situation: have mums performed outside work? What digital devices have caused the most conflicts in modern families?

 Is it possible to answer these questions?

One of the advantages of the article is the attempt to indicate the possibilities of supporting the family also in this type of conflict.
